# TOKEN ALIGNMENT VIA CHARACTER MATCHING FOR SUBWORD COMPLETION

## ABSTRACT

Generative models, widely utilized in various applications, can often struggle with prompts corresponding to partial tokens. This struggle stems from tokenization, where partial tokens fall out of distribution during inference, leading to incorrect or nonsensical outputs. This paper examines a technique to alleviate the tokenization artifact on text completion in generative models, maintaining performance even in regular non-subword cases. The method, termed *token alignment*, involves backtracking to the last complete tokens and ensuring the model's generation aligns with the prompt. This approach showcases marked improvement across many partial token scenarios, including nuanced cases like space-prefix and partial indentation, with only a minor time increase. The technique and analysis detailed in this paper contribute to the continuous advancement of generative models in handling partial inputs, bearing relevance for applications like code completion and text autocompletion.

## 1 INTRODUCTION

Generative models have shown remarkable efficacy in a range of applications. However, they have been observed to falter when dealing with partially provided inputs or subwords during text *completion*. For instance, a generative model might struggle to predict the remaining part of the word where a prompt ending in a subword often leads to incorrect or nonsensical outputs. This issue arises due to the artifact of tokenization where a partial token can be out-of-distribution during inference. In this paper, we introduce a method to address and rectify these shortcomings in generative models, ensuring they maintain their performance on complete contexts.

Our approach involves backtracking to the last complete tokens and aligning the model's generation to match with the given prefix, as shown in Figure 1. For example, if the end of the prefix is "sys" we guide the model's generation to ensure compatibility with this prefix where all possible tokens match with "sys". This method not only improves the accuracy of the model's predictions on subword data metrics but also retains the performance on generic evaluation sets.

To illustrate our approach, consider the following programming code snippet where the end of the prompt is right after the incomplete token "sys" in Figure 2. A conventional model might generate an incorrect or suboptimal output such as "sysYtem", while the correct prediction should be "system". Our approach solves this issue by matching the characters starting from the last complete pre-token and aligning the subsequent tokens with the prefix. The results of our approach, as shown in various examples, demonstrate the model's ability to reliably generate the correct output regardless of the partial context.

Our evaluation of the token alignment approach indicates noticeable improvement in handling partial token contexts, outperforming the traditional decoding approach without any subword handling. We present comprehensive evaluation results on many partial token scenarios such as subword of natural words, as well as much less obvious cases of partial tokens related to punctuations, or artifacts of contiguous white spaces and space prefix. Our approach also does not incur much extra latency, with an average increase of only 3-7 ms for using token alignment, in addition to the number of backtracked tokens, thanks to our implementation of an efficient trie-based lookup table. We believe our findings provide a meaningful and practical contribution to the ongoing work in improving robustness of generative models, especially related to handling partial inputs.

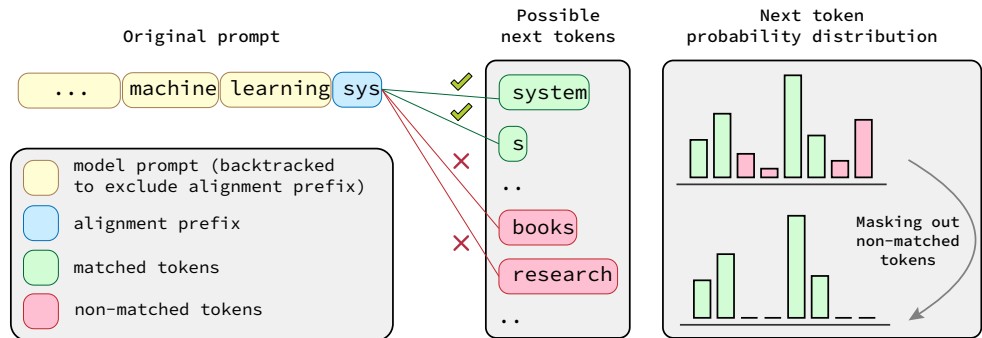

Figure 1: An illustration of token alignment process. We first tokenize the prompt where the last tokens may correspond to partial tokens. We use the backtracked prompt as model input and use the alignment prefix to filter possible tokens at byte or character level. We then mask out non-matched tokens by zeroing their probabilities, which is later used for sampling to select the next token.

```
# write a function to get three maximum
    numbers from a list
def three_max(l):
    re<T> = []
    for i in range(len(l)):
        if i == 0:
            re.append(l[0])
```

(a) Without token alignment

```
# write a function to get three maximum
    numbers from a list
def three_max(l):
    re<T>turn sorted(l, reverse=True)[:3]
```

(b) With token alignment

Figure 2: Effects of token alignment on prompts ending with a subword. The `<T>` marks the end of the prompt, after which the completion is shown. Without token alignment, the model fails to predict "return" correctly, as the sequence of tokens "re", "turn" is out-of-distribution. Token alignment alleviates this constraint by backtracking to full tokens before "re", then align subsequent generations with the prompt.

## 2 RELATED WORK

Recent works have identified severe robustness issues for large language models. For instance, simple perturbations like synonym substitution on one word can significantly change outputs and fool the models (Jin et al., 2020; Zang et al., 2020). Most studies focus on general perturbations especially at random locations of prompts for both text (Dhole et al., 2021; Nie et al., 2020; Wang et al., 2021; Kiela et al., 2021) and code models (Wang et al., 2023; Li et al., 2022; Yang et al., 2022; Jha & Reddy, 2023). However, we see a large gap of partial versus full tokens at the end of prompts with only a few characters missed or added (e.g., with or without a white space) due to the artifact of tokenization. This robustness problem is barely investigated. Subword regularization tried to partially mitigate such problems by randomly introducing sub-tokens in training and improve robustness behavior (Kudo, 2018a). However, this requires to retrain the models and also leads to inference latency increase. Our approach, on the other hand, can work for any models with negligible latency increase.

We acknowledge concurrent work in the form of a blog post namely token healing (Lundberg, 2023), which attempts to similarly solve the partial token problems on inference time. However, we independently develop our own token alignment algorithm in a more general design. We also thoroughly validate and analyze the effectiveness of token alignment with extensive empirical results on various tasks and many partial token scenarios.

## 3 METHODOLOGY

Generative language models have been remarkably successful in a variety of applications, including code completion in Integrated Development Environments (IDEs) and text autocompletion in email or productivity tools. However, these models often encounter difficulties when dealing with partial

or incomplete inputs, which we refer to as the partial token issue. In such cases, the models tend to generate outputs that are not compatible with the given context due to the constraint imposed by tokenization. This limitation significantly hampers the utility and user experience of applications built on these models. To tackle this issue, we propose a method called token alignment which makes use of a character or byte level trie for efficient matching along with masking cache to increase efficiency further. The underlying principle of our approach is to backtrack to the last complete token and constrain the model's generation in a way that aligns with the given prefix.

In our methodology, we first define the last complete token by examining the tokenization of the input sequence. Given the tokenization of the last line as $T_1, T_2, ..., T_n$, we start generating from the $-B^{th}$ token, i.e., $T_{n-B+1}$, where $N$ is the number of tokens we need to backtrack. The rest of the context is the sequence $T_{n-B+1}, ..., T_n$, which we refer to as the *alignment prefix* where the subsequent generation needs to match at a byte level (or character level if the tokenizer is not a byte-level BPE).* The token alignment step ensures that the next token starts with the alignment prefix or is a prefix of the alignment prefix itself by masking out probabilities of tokens that do not match. After each step where the next token is selected, we deduct the generated bytes from the alignment prefix until the alignment prefix is empty, after which we finish the token alignment process.

---

**Algorithm 1** TokenAlign Algorithm

---

1: **procedure** TOKENALIGN($T, N$)
2:   **Define:**
3:     $X$: The input tokens $[x_1, x_2, ..., x_n]$
4:     $B$: The number of tokens to backtrack
5:     $C$: Tokens for model context
6:     $V$: Prebuilt character or byte-trie
7:     $P$: Alignment prefix
8:     $p$: Probability distribution of next token
9:     $x'$: Selected next token
10:   $C \leftarrow X[:-B]$
11:   $P \leftarrow join(X[-B:])$
12:   **while** $P$ is not empty **do**
13:     $p \leftarrow model(C)$
14:     $T \leftarrow V(P)$ (tokens compatible with $P$)
15:     $p_{new}(T) \leftarrow 0$
16:     $x' \sim p_{new}$ (sample)
17:     $P \leftarrow P[len(x'):]$
18:     $C$.append($x'$)
19:   **end while**
20:   **return** $C$
21: **end procedure**

---

The efficiency of our approach is supported by two key techniques: a character-trie for fast prefix matching, and a mask cache. The character-trie is a tree data structure, where each node represents a character and the paths from the root node to the leaf nodes represent complete tokens in the vocabulary. We pre-build this trie using the model's vocabulary before starting the generation process, thereby enabling a fast and space-efficient representation of the vocabulary.

The mask cache is used to further accelerate the lookup process to avoid unnecessary the trie lookup. For common cases, such as a single space as context, we pre-build and cache a Boolean mask. This cached mask is then used for quickly filtering tokens that share a common prefix with the given context. These techniques significantly reduce the latency, enabling our method to be highly efficient. The use of a trie-based lookup table and the pre-built mask cache results in a time saving compared to a naive implementation using a dictionary lookup. In sum, our methodology presents an effective and efficient solution to the partial token issue in generative models, enhancing their performance in applications such as code completion and text autocompletion.

## 4   PARTIAL TOKEN SCENARIOS

We give an overview of many partial token scenarios where the model is susceptible to degenerate behavior when the prompt ends with such partial token. Such partial token scenarios range from obvious cases such as natural subwords to less obvious cases such as space prefix, as illustrated in Figure 3.

  1. **Subword**. This case refers to natural language subword. For instance, a possible subword of "banana" is 'banan' or 'bana'. We note that while it is possible for a tokenizer to use a

---

*We use character of byte interchangeably while describing the tokenizer.

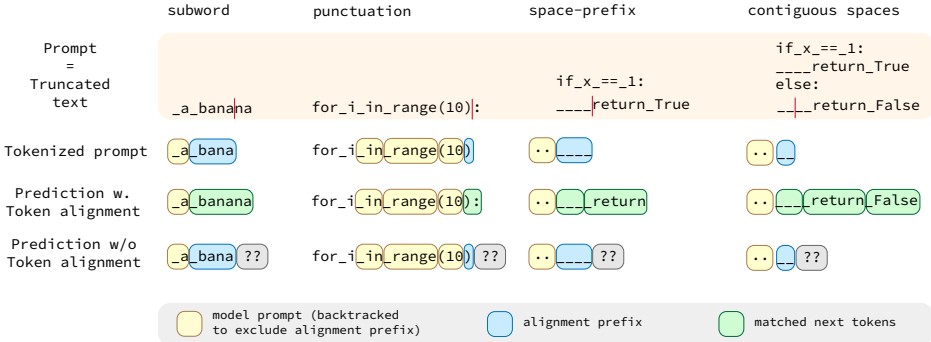

Figure 3: Partial token scenarios. Note: the underscore denotes a space character.

subword as an actual token, such as in the case of "beautifully" which can be represented as 'beautiful' + 'ly', not all possible subwords will correspond to a token, e.g., 'beautifu' is unlikely to be a full token.

2. **Punctuation**. It is possible that multiple punctuations are grouped together as a token during the tokenizer training process, which can lead to some punctuations being a partial token in a non-obvious way. For example, in Python, it is possible that the code snippet `if (x==1)` ends in a partial token ')' of a full token '):', which can lead to a suboptimal completion behavior if presented to a model. In the case of punctuation, it is general quite hard to judge which case would correspond to a partial punctuation. We separate out this case to emphasize the importance of token alignment as a universal method that can handle both prompt ending in either partial token or non partial token in a prompt-agnostic way.

3. **Space prefix.** Most tokenizers use the space-prefix schema where a whitespace is often grouped as part of a word as a prefix (Scao et al., 2022; Brown et al., 2020; Li et al., 2023; OpenAI, 2023). This schema is primarily aimed to reduce the number of tokens required to represent text. For example, with space-prefix, 'I_like' can be tokenized into two tokens, 'I' and '_like', instead of three ('I', '_', and 'like'). If a prompt 'I_' is presented to the model, such model can have difficulty predicting coherent text since the prompt is a sequence of tokens ('I', '_'), which is out-of-distribution compared to the training stage which observes ('I', '_like').

   The space prefix's constraint is also quite pronounced in the presence surrounding indentation block. For instance, ____x= is tokenized into ('____', '_x', '='). However, if the prompt ends after the full block of indentation ____, this becomes a constraint since ('____', 'x') is out-of-distribution compared to ('___', '_x') seen during training. We refer to this as the *space prefix with indentation* case.

4. **Contiguous Spaces**. Modern tokenizers often intentionally group whitespaces together for improved compression rates. For instance, Codex (Chen et al., 2021) extends the GPT-3 tokenizer (Brown et al., 2020) to include contiguous spaces of different lengths (up to 65 contiguous spaces in some version of TikToken (OpenAI, 2023), for instance). Other models such as LLaMA (Touvron et al., 2023), StarCoder (Li et al., 2023), CodeGen (Nijkamp et al., 2022), to name a few, also use such contiguous white spaces.

   While such grouping of white spaces is compression efficient, it can lead to out-of-distribution behavior if the prompt ends in partial token of such contiguous white spaces. For instance, consider the following prompt '__if True:\n__' where _ represents a space character. The correct syntax for Python is such that what comes after the if clause requires another level of indentation, meaning that the model should predict extra _ characters. However, the model will have a hard time completing the next two _ since during training, the model always obvious contiguous ___if it were to complete the next level of indentation as in this example.

## 5 EVALUATION

In this section, we empirically demonstrate that each of the partial token scenarios described in Section 4 can unnecessarily constrain the model due to the artifact of tokenization and lead to significant drops in evaluation scores. We perform the evaluation using both code generation tasks where partial token completion such as in the use of code-completion tools can lead to issues if not well handled, as well as natural language tasks such as text completion and natural language understanding. We construct the evaluation datasets for each case described in Section 4 show that token alignment method graciously handles the constrain due to all such cases, making a universal partial token handling approach for language model inference.

### 5.1 DATASETS

To demonstrate the sensitivity of language models on how the prompt ends, we analyze language models' behavior by processing publicly available datasets to their corresponding variants, in order to isolate the effects of each case. Below, we detail our methods of processing evaluation datasets based on public benchmarks for both code generation and natural language tasks.

#### 5.1.1 CODE GENERATION TASKS

**Execution-Based Code Generation Benchmarks**   We assess code generation abilities using the MBXP benchmark (Athiwaratkun et al., 2022), which is a multi-lingual version of MBPP (Austin et al., 2021), focusing on datasets in Python, Java, and JavaScript. This entails evaluating function completion from partially given canonical solutions.

For each dataset: (1) the prompt is formed by merging the original function signature with a section of the canonical solution, and (2) this section is selectively cut at a specific location based on different criteria depending on the scenarios.

- **Subword:** The cut is based on space delineation, yielding prompts ending in subwords.
- **Punctuation:** Cuts are made within punctuation sequences to demonstrate challenges with subtokens. For instance, '`{};`' might be cut to produce '`{`' or '`{}`'.
- **Space prefix with indentation:** Here, we focus on challenges arising from prompts ending with spaces used for indentation. For instance, in a snippet like `\n␣␣␣␣return␣value`, a cut after the indentation spaces produces `\n␣␣␣␣`. Such endings can lead to suboptimal generation behavior without token alignment due to the last space corresponds to a space prefix of a future word such as in `␣return`.
- **Space prefix without indentation:** Cuts are made between spaces and non-spaces while avoiding spaces specifically used for indentation. For example, from the snippet `\n␣␣␣␣return␣value`, a possible cut could yield `\n␣␣␣␣return␣`.
- **Contiguous spaces:** Cuts are introduced within sequences of contiguous whitespace characters. Here, we consider `␣`, `\n` and `\t` as candidates for whitespace.

Each dataset also has a **baseline** where prompts are backtracked to end at the last full word, allowing us to evaluate model performance without the challenge of partial tokens. For example, in the subword case, if the prompt ends with '`for␣i␣in␣rang`', we truncate it to be '`for␣i␣in`'. In the baseline scenario, models can perform well without the use of token alignment since it does not correspond to an ending partial token. The evaluation scores should ideally be slightly lower due to *lower amount of information* in the truncated prompt. The goal of the corresponding baseline evaluation datasets is such that the prompt should not end in partial token, so that we can use them (1) to see that token alignment does not degrade performance for prompt ending with non partial token and (2) measure improvement on the partial token dataset due to token alignment.

**Metrics**   We use pass@k (Kulal et al., 2019) with an unbiased estimate by Chen et al. (2021) as the execution-based evaluation metric to measure the functional correctness of the generated code snippet. Execution-based metrics is robust to variation in the generated code and can better reflect the actual problem solving abilities of language models, which is highly suitable for our purposes as an approach to test the effects of token alignment.

Table 1: **Token alignment for subword scenario.** This table illustrates pass@1 scores (%) on MBXP partial benchmark as well as subword version of SQuAD and Wikitext, which show clear improvement due to token alignment.

| | Token Alignment | Subword MBXP | | | Baseline of Subword MBXP | | |
|---|---|---|---|---|---|---|---|
| | | Python | Java | JavaScript | Python | Java | JavaScript |
| StarCoder | with | **56.58** | **52.17** | **49.31** | **54.32** | 49.54 | 50.16 |
| | w/o | 30.25 | 25.40 | 30.74 | 53.40 | **51.37** | **51.12** |
| LLaMA-7b | with | **27.47** | **20.82** | **24.87** | 23.66 | **20.02** | 26.36 |
| | w/o | 13.48 | 10.76 | 17.50 | **25.82** | 19.68 | **26.68** |

| | Token Alignment | Subword SQuAD | | Original SQuAD | |
|---|---|---|---|---|---|
| | | EM | ES | EM | ES |
| LLaMa-7b | with | **40.27** | **78.81** | **14.65** | **52.24** |
| | w/o | 12.42 | 69.49 | 6.04 | 48.44 |

| | Token Alignment | Subword WikiText | | | Baseline of Subword WikiText | | |
|---|---|---|---|---|---|---|---|
| | | Acc. | ES | Rouge-L | Acc. | ES | Rouge-L |
| LLaMa-7b | with | **19.42** | **47.80** | **0.257** | **1.45** | 42.826 | 0.153 |
| | w/o | 10.9 | 44.50 | 0.190 | 0.75 | **44.141** | **0.168** |

### 5.1.2 NATURAL LANGUAGE TASKS

We adapt several natural language evaluation benchmarks to reflect the impact of the subword in the prompt.

- Question answering: SQuAD (Rajpurkar et al., 2016; 2018) with exact match metric and edit similarity metric.
- Text generation: Wikitext (Merity et al., 2016) with first token accuracy and first $n^{th}$ word fuzzy-matching based metrics.

For all datasets, we use the original context and part of the ground truth as our prompt, where the prompt ends with subword or without (control version). The control version always has one space-delineated word fewer than the subword counterpart. In terms of partial token scenarios for text-based evaluation, we primarily consider the subword and space prefix.

**Metrics** For question answering tasks, the ground-truth is a list of answers where each answer can be one or more words. We use the edit similarity and exact match as the metrics. For text generation with wikitext, we found that exact match metric is too strict for this type of open-ended generations in open domains. Therefore, in addition to the first token accuracy, we also use fuzzy-matching based metrics to calculate the similarity between the first $n^{th}$ word of the generation and the groundtruth with $n = 50$ in our case. Here we use the edit similarity and Rouge-L (Lin, 2004) as the metrics.

### 5.2 TOKEN ALIGNMENT HANDLES MANY PARTIAL TOKEN SCENARIOS

In this section, we show that without token alignment, publicly available language models can suffer from the partial token issue during text completion. We show that token alignment offers a simple solution that applies to all the prompt scenarios considered. We primarily use StarCoder (Li et al., 2023) and LLaMA 7B (Touvron et al., 2023) to perform this study.

### 5.2.1 SUBWORD

**Code Generation** The results with and without token alignment in Table 1 demonstrate clear differences in the ability to handle prompts that ends with subword where the performance can drop by $\approx 14 - 22\%$ for pass@1 without token alignment. For all the results, we **boldface** the scenario that obtain higher performance and highlight in red the scenario where the different is greater than

Table 2: Token alignment for partial token of **punctuation** scenario.

| | Token Alignment | Punc MBXP | | | Baseline of Punc MBXP | | |
|---|---|---|---|---|---|---|---|
| | | Python | Java | JavaScript | Python | Java | JavaScript |
| StarCoder | with | **59.35** | **48.80** | **46.03** | **58.31** | **51.95** | **46.41** |
| | w/o | 45.10 | 31.78 | 24.23 | 58.31 | 49.31 | 46.15 |
| LLaMA-7b | with | **31.75** | **20.93** | 26.79 | **31.90** | 21.44 | 27.05 |
| | w/o | 22.11 | 11.85 | 14.87 | 31.45 | **24.21** | **27.95** |

4 absolute points for pass@1. We also show the scenarios where the prompt ends with a full word (space-delineated) as a control experiment. In this case, we can see that token alignment yields similar scores as without token alignment, indicating that token alignment can be used regardless of whether the prompt ends with or without a partial token.

**Natural Language Tasks** We also show the text evaluation results in Table 1 where we observe clear performance gain from token alignment. Observe that in the case of question answering (SQuAD), even though we provide partial information in terms of subword, akin to providing a hint, most models fail at answering correctly due to the constraint from the partial token artifact. This resulting in subword SQuAD scores being lower than the baseline SQuAD even though the subword dataset contains extra information in the prompt. However, once we use token alignment which alleviates the tokenization constraint, the extra information helps significantly, allowing many models to score in the range of $40\%$ exact match versus $15\%$ for the baseline case.

For text generation on WikiText, the next word accuracy, edit similarity, and ROUGE scores portray obvious trends. When the prompt ends with a subword, all scores drop noticeably compared to the baseline. Token alignment helps remove the tokenization constraint, resulting in significant improvement for all metrics. On the control datasets which differ from the original datasets by one fewer (sub)word, using token alignment leads to similar or slightly better results. The finding indicates that token alignment can be use in all cases without the need to detect whether the prompt ends with a subword or not.

### 5.2.2 PUNCTUATION

Table 2 demonstrates the results from the punctuation MBXP dataset, highlighting a notable improvement due to the use of token alignment. As an example, consider a prompt `if x=` representing a code snippet in Python. It is unambiguous that the next character should be $=$ since otherwise the Python syntax would not be correct. However, this prompt corresponds to an ending partial token $=$ of a complete token $==$; therefore, predicting an extra token `'='` is out-of-distribution compared to what is seen during training, which is `'=='`. This dynamic is distinct from the natural subword scenario where boundaries are determined by whitespace and the identification of partial tokens is more clear. When punctuation is involved, avoiding an ending with a partial token becomes a nuanced challenge, which token alignment effectively addresses.

### 5.2.3 SPACE PREFIX

The results from the space prefix MBXP datasets (for code generation) and space-SQuAD are shown in Table 3, illustrating that a presence of a extra trailing token, while seemingly innocuous, can noticeably degrade generation behavior without token alignment. For the code generation task, we also have a dataset split *prefix-indent* to ablate the case of prefix space with indentation. Table 3 illustrate the results where we observe slight improvement in the *prefix-sep* case for code and a strong improvement for text on SQuAD. We provide additional observations in Appendix A.2.2.

### 5.2.4 CONTIGUOUS SPACES

The results on the contiguous space MBXP dataset in Table 4 show that token alignment alleviates the partial token issue significantly, especially in Python, where the syntax is most sensitive to spaces. We emphasize that, superficially, this case does not seem obviously linked to partial token issues (see

Table 3: **Space prefix** is a common design approach for many modern tokenizers for efficient compression, but can lead to suboptimal generation if the prompt ends with a space character (due to out-of-distribution). Token alignment resolves such problem gracefully, reflected by improved pass@1 scores (%) on prefix-sep MBXP and prefix-indent MBXP. FS denotes few-shot prompting.

| | Token Alignment | Prefix-sep MBXP | | | Baseline's | | |
|---|---|---|---|---|---|---|---|
| | | Python | Java | JavaScript | Python | Java | JavaScript |
| StarCoder | with | **56.09** | **50.75** | **48.16** | 56.73 | **50.64** | **48.83** |
| | w/o | 54.06 | 48.67 | 44.02 | **57.48** | 49.94 | 46.93 |
| LLaMA-7b | with | **25.85** | **21.67** | **25.92** | 23.82 | **22.36** | 25.59 |
| | w/o | 22.86 | 20.74 | 23.35 | **25.00** | 21.55 | **26.48** |

| | Token Alignment | Prefix-indent MBXP | | | Baseline's | | |
|---|---|---|---|---|---|---|---|
| | | Python | Java | JavaScript | Python | Java | JavaScript |
| StarCoder | with | **42.95** | **50.19** | **49.13** | 48.82 | **50.19** | 46.89 |
| | w/o | 17.10 | 49.31 | 45.33 | **49.02** | 50.06 | **47.75** |
| StarCoder FS | with | **45.52** | **50.44** | **46.89** | 52.94 | **51.57** | **46.37** |
| | w/o | 24.51 | 48.93 | 42.56 | **53.14** | 52.07 | 46.02 |
| LLaMA-7b | with | **15.45** | 20.45 | **28.03** | 17.30 | 22.21 | **25.95** |
| | w/o | 12.05 | **20.95** | 24.39 | **18.02** | **22.33** | 25.43 |
| LLaMA-7b FS | with | **17.92** | **23.59** | **25.26** | 18.74 | 20.83 | **26.30** |
| | w/o | 15.24 | 21.83 | 24.74 | **19.36** | **21.46** | 25.78 |

| | Token Alignment | Space Prefix SQuAD | | Baseline's | |
|---|---|---|---|---|---|
| | | EM | ES | EM | ES |
| LLaMA-7b | with | **26.82** | **64.34** | **9.51** | **62.22** |
| | w/o | 20.45 | 61.34 | 4.38 | 61.88 |

Figure 3). Hence, we separated it to highlight its importance and to measure the effects with and without token alignment.

Besides this processed dataset, token alignment also impacts the *regular* execution-based evaluation significantly for some models that employ aggressive white space grouping in their tokenizer for high compression. For example, in the StarCoder model, a newline character is often grouped with spaces, such as \n␣␣␣␣, leading prompts ending with a newline character, \n, to correspond to a partial token case. Without token alignment, we observe suboptimal behavior, as illustrated in Table 4, where the execution scores drop from $44.0\%$ to $27.2\%$ without token alignment for MBXP JavaScript. For instance, when the model encounters a JavaScript prompt ending with {\n, indicating the beginning of a function, it might generate }\n without actually completing the function. Similar observations occur in other programming language benchmarks.

## 6 ABLATION STUDY AND GENERALITY OF TOKEN ALIGNMENT

In this section, we also discuss an alternative approach to subword completion by processing training data by subword regularization. We show that our token alignment method is complementary and can offer a robust approach to handle subwords compatible with subword regularization. We also discuss latency as well as ablation study on the number of backtrack tokens.

### 6.1 COMPARISON AND COMPATIBILITY WITH SUBWORD REGULARIZATION

Subword regularization performs tokenization in a more random manner where each word can be tokenized differently without a fixed pattern. We trained a model with subword regularization on top of a regular language model and compare with the performance with base model before subword regularization, with and without token alignment (details in Appendix A.2.1). We find subword regularization alone can perform quite well on many partial token scenarios, almost matching the scores obtained from token alignment (Table 5). Token alignment helps increase the scores further,

Table 4: Evaluation scores (pass@1 %) for models on the processed contiguous space MBXP and regular MBXP datasets, considering the influence of token alignment and the partial token constraint due to **contiguous spaces** in the tokenization.

| | Token Alignment | Contiguous Space MBXP | | | Baseline's | | |
|---|---|---|---|---|---|---|---|
| | | Python | Java | JavaScript | Python | Java | JavaScript |
| StarCoder | with | **43.93** | 49.20 | **49.41** | 47.94 | 48.86 | **50.37** |
| | w/o | 32.00 | **50.46** | 48.45 | **49.28** | **50.92** | 49.73 |
| LLaMA-7b | with | **16.56** | 18.19 | 23.91 | 15.64 | 17.39 | 23.05 |
| | w/o | 14.20 | **19.11** | **24.01** | **18.72** | **18.99** | **24.44** |

| Token Alignment | MBXP with StarCoder | | | | | | | | |
|---|---|---|---|---|---|---|---|---|---|
| | Python | JS | Java | C++ | Swift | TS | Kotlin | Go | Scala |
| with | **42.5** | **44.0** | **43.0** | **43.3** | **27.6** | **42.3** | **37.9** | 30.8 | **34.1** |
| w/o | 7.5 | 27.2 | 42.0 | 39.2 | 8.9 | 7.0 | 16.4 | 30.8 | 22.4 |

indicating that subword regularization itself can benefit from token alignment process. Overall, we find that token alignment can help alleviate constraints on tenuous cases that subword regularization does not address. We provide detailed results in Appendix A.2.1.

## 6.2 LATENCY AND THE NUMBER OF BACKTRACK TOKENS

The extra latency from token alignment comes mainly from two components. First, backtracking results in more tokens needed during incremental decoding. For the usual decoding process, the latency added is $B' \cdot \ell_d$, where $B'$ is the number of decoding steps to finish the character or byte matching process of token alignment, and $\ell_d$ is the incremental decoding latency per token.[†] We show the distribution of the number of steps $B'$ is Figure 4 in Appendix, which shows that $B'$ peaks at the number of backtrack tokens $B$ and can be slightly lower or higher. Second, there is a minimal cost of performing alignment $\ell_{TA}$, which entails trie lookup and masking the next token probabilities, which is in the order of $< 1$ ms in most cases. After the alignment prefix (Figure 1) is fully matched and becomes an empty string, we stop the alignment process and no longer incurs the extra latency per token $\ell_{TA}$. Overall, the minimal extra latency makes it viable to use this algorithm in real time text or code completion applications.

Throughout this experiment, we use a fixed number of tokens to backtrack $B = 3$ and provide an ablation study for the importance of this backtrack tokens. The detailed result in Appendix Table 6 suggests that $B = 1$ performs slightly worse than $B = 3$ with a difference of $-3.7\%$ on average. Since the latency for the $B = 3$ is not much more than that of $B = 1$, we use $B = 3$ as the default setting. We note that some tokenizers use pre-token schema which offers opportunities to mark some token as beginning of pretoken as building time, such as in SentencePiece (Kudo & Richardson, 2018), in which case we can dynamically determine the number of backtrack tokens based on the context. However, most tokenizers do not have this feature so we use a fixed $B$.

## 7 CONCLUSION

This paper explored the complex challenge of handling partial tokens in generative models, introducing specific scenarios such as subword, punctuation, space prefix, and contiguous spaces. Through the application of token alignment, we demonstrated noticeable improvements across these scenarios. In the future, as tokens become longer due to possible advancement in tokenizer compression, the token alignment may become increasingly important as more cases may subtly fall into the categories of partial tokens. The findings of this study emphasize the importance of understanding tokenization artifacts and present a meaningful advancement in enhancing the robustness and practical applicability of generative models in various real-world tasks.

---

[†]With speculative decoding, it will have a speedup due to lower amortized latency per token.

## 8 LIMITATIONS

While token alignment works in principles on any tokenizer, it may require additional attention to details to implement, especially with tokenizers that are not lossless which could make the token alignment process tricky. The latency addition due to token alignment is generally small for long generation, but can be a larger portion of overall budget if the generation is short.

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

## A    APPENDIX

### A.1    FAQs

1. **Q**: Is token alignment absolutely necessary?
   **A**: Yes and no. In the scenario of text *completion* such as the scenarios in IDE environment, token alignment is highly beneficial. In the formation of question answering in chat styles (such as in ChatGPT), the model can typically generate the completion without being constrained by the artifact of tokenization.

2. **Q**: Is token alignment compatible with the code insertion scenario?
   **A**: Yes. Since the insert scenario (Bavarian et al., 2022; Fried et al., 2022) performs text *completion* where the prompt during inference uses the right context follows by left context, the insertion is simply a continuation. This makes the token alignment method compatible with the insertion scenario.

3. **Q**: Is it compatible with speculative decoding?
   **A**: Yes. For speculative decoding (Leviathan et al., 2022; Chen et al., 2023), the draft model can propose tokens that match the prompt during token alignment, which will be either accepted or rejected by the main model based on the token sequence.

### A.2    ADDITIONAL RESULTS

#### A.2.1    SUBWORD REGULARIZATION

In Table 5, we outline the results where we show the performance of a model trained without subword regularization (the usual training strategy employed in many language models) and with additional training with subword regularization (Kudo, 2018b). We use the implementation given in SentencePiece (Kudo & Richardson, 2018) with $\alpha = 0.05$. For instance, `New York` is segmented into either `N, e, w, _York` or `New, _York`, where lower alpha means higher chances of the canonical segmentation being more common. The results indicate that subword regularization without token alignment indeed helps increase the performance in the subword MBXP from 6.89% to 18.11%, but using token alignment with subword regularization model increases the scores further to 18.52%.

This finding highlights the generality of token alignment where it can be used with or without subword regularization, and can even help address cases where subword regularization alone may not handle based on the consistent improved scores of token alignment. In addition, subword regularization may introduce additional latency cost due to finer grain tokenization of text in the training stage, which influences how the tokens are generated during inference. We leave this direction of research on investigating the interplay of subword regularization and tradeoff for future work.

#### A.2.2    ADDITIONAL OBSERVATIONS

Note that on the *prefix-indent* split, with token alignment, the generation's quality improved significantly but not quite matching the baseline (e.g. 42.95 for partial token + token alignment versus 48.82 for baseline for Python StarCoder). We found that the subpar score compared to baseline is due some examples corresponding to a function signature and extra spaces, which can confuse the model that is not fully preference aligned. We include the few-shot prompting version (FS) which includes examples of complete functions prepended at the beginning, which is to help steer the model

Table 5: Effects of subword regularization and token alignment

| Model | Token Alignment | Partial Word MBPP | Baseline Word MBPP | Partial Punc MBPP | Baseline Punc MBPP |
|---|---|---|---|---|---|
| Subword | with | **18.52%** | 15.43% | 17.96% | **19.44%** |
| Regularized | w/o | 18.11% | **17.39%** | **18.10%** | 19.14% |
| Baseline | with | **16.67%** | 12.96% | **18.84%** | 18.25% |
| 600M | w/o | 6.89% | 13.89% | 11.72% | **18.69%** |

Table 6: Execution-based evaluation on various partial token versions of MBXP (pass@1). The difference $\Delta$ denotes the scores of B=-1 minus the scores of B=3. The total overall difference is -3.69%.

| Token Alignment | Partial Word MBXP | | | Baseline's | | |
|---|---|---|---|---|---|---|
| | Python | Java | JavaScript | Python | Java | JavaScript |
| B=1 | 56.58% | 52.17% | 49.31% | 54.32% | 49.54% | 50.16% |
| B=3 | 54.01% | 47.37% | 43.76% | 54.12% | 52.29% | 50.59% |
| $\Delta$ | 2.57% | 4.81% | 5.55% | 0.21% | -2.75% | -0.43% |
| | Partial Punc MBXP | | | Baseline's | | |
| B=1 | 59.35% | 48.80% | 46.03% | 58.31% | 51.95% | 46.41% |
| B=3 | 59.64% | 50.82% | 47.44% | 58.90% | 50.95% | 44.74% |
| $\Delta$ | -0.30% | -2.02% | -1.41% | -0.59% | 1.01% | 1.67% |
| | Prefix-sep MBXP | | | Baseline's | | |
| B=1 | 56.09% | 50.75% | 48.16% | 56.73% | 50.64% | 48.83% |
| B=3 | 58.23% | 51.22% | 48.16% | 57.48% | 50.17% | 47.71% |
| $\Delta$ | -2.14% | -0.46% | 0.00% | -0.75% | 0.46% | 1.12% |
| | Prefix-indent MBXP | | | Baseline's | | |
| B=1 | 42.95% | 50.19% | 49.13% | 48.82% | 50.19% | 46.89% |
| B=3 | 44.08% | 50.31% | 46.71% | 49.54% | 53.32% | 47.23% |
| $\Delta$ | -1.13% | -0.13% | 2.42% | -0.72% | -3.14% | -0.35% |
| | Contiguous Space MBXP | | | Baseline's | | |
| B=1 | 43.93% | 49.20% | 49.41% | 47.94% | 48.86% | 50.37% |
| B=3 | 45.88% | 50.23% | 48.99% | 49.38% | 49.66% | 49.73% |
| $\Delta$ | -1.95% | -1.03% | 0.43% | -1.44% | -0.80% | 0.64% |

to complete the function and observe noticeable improvement, but still does not quite close the gap. We hypothesize with a model that is more preference aligned will handle such cases better, where this partial token dataset can be used as once of such benchmark to measure instruction following abilities.

### A.2.3 NUMBER OF BACKTRACK TOKENS

We show detailed results on evaluating with varying number of backtrack tokens $B$ in Table 6. While the results for individual datasets vary, on average, the overall difference between the scores from $B = 1$ and $B = 3$ is $-3.69\%$, indicating that $B = 1$ is less accurate since it may not backtrack enough to a full token.

In terms of latency, the factor that affects the latency the most is the number of steps it takes to get out of the character matching mode during token alignment. Figure 4 shows the distribution of the number of steps in the case of $B = 3$, where the distribution peaks around $B$ steps, with some long tail towards higher the number of steps. We emphasize that the number of steps required to complete the character matching need not be exactly equal to the number of backtrack tokens. This is because given a string that corresponds to the backtracked prompt, the model can generate any tokens as long as it matches, which could result in breaking the string down to coarser or finer granularity.

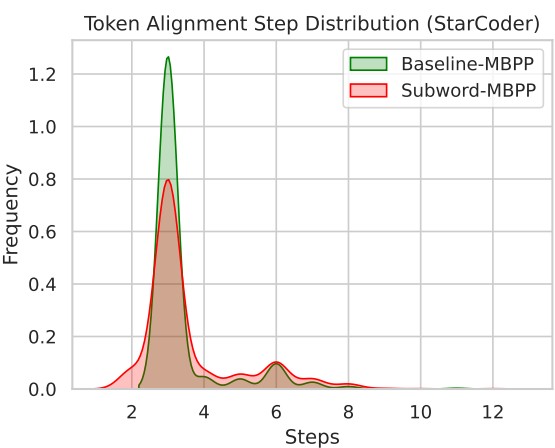

Figure 4: The density estimation of the number of steps during matching process in token alignment.

