# OpenReview forum: "Token Alignment via Character Matching for Subword Completion"
_ICLR.cc/2024/Conference — Submitted to ICLR 2024_

### Official Review · Reviewer_jqrQ · 2023-10-30

**Soundness:** 2 fair
**Presentation:** 2 fair
**Contribution:** 2 fair
**Rating:** 5
**Confidence:** 4

**Summary:**

This paper explored handling partially labeled complex tasks in generative models. Through token alignment, improvements in various scenarios were demonstrated.

**Strengths:**

The proposed token alignment method could be combined with multiple techniques, such as subword regularization.

The method presented was meaningful for code completion and text output directions.

**Weaknesses:**

In terms of prefix-indentation splitting, the results of the proposed method still lagged behind the baseline.

The method introduced a certain time delay.

Prompts affected the results.

**Questions:**

1. The outputs of LLMs were uncertain. Even a minor change in a prompt could lead to variations in the output. During the use of prompts in the paper, was the specific impact of the prompt considered?

2. Given the powerful In-Context Learning capabilities of large language models, it would be worth exploring whether adding relevant knowledge to the prompt could further enhance the proposed method.

3. Was there any consideration that an excessive amount of code data in the dataset might dilute the pre-trained knowledge in LLMs, impairing code generation capabilities?

4. How did the proposed method perform in multilingual or cross-lingual scenarios?

5. The paper mentioned "token healing." Could there be a detailed comparison between "token alignment" and "token healing" in terms of performance across different datasets and application scenarios? What were the pros and cons of these two methods when dealing with partially labeled issues?

6. In which areas did "token alignment" excel? Were there scenarios or applications where other methods might be more appropriate?

7. For long texts or texts exceeding the model's maximum input length, did the "token alignment" method maintain its effectiveness? In such cases, was there a need to adjust or optimize the method?

---

> ### Author Response · Authors · 2023-11-23
> **Response to Reviewer jqrQ**
>
> Thank you for the reviewer for your comment and thoughtful questions.
>
>
> Addressing questions:
> 1. This is a good point. We use the baseline datasets where we pre-backtrack so that the prompt does not contain partial token. Therefore, the difference between the baseline and non baseline is as low as possible, in order to control the effect of the prompt. Moreover, we average the results over many curated datasets to understand the aggregate behavior.
>
> 2. Good point. One way to use in-context learning is to make the model repeat the entire input and generate full tokens that match the partial tokens instead. However, if the input is long, this will amount to high extra latency. Therefore, we do not explicitly consider this approach.
>
> It is possible that we do in-context learning in such a way that makes the model output only a snippet shortly before the end of the prompt. Such method is likely quite complicated and requires processing to extract out the right portion.
>
> Our method presents a clean way to align tokens natively by guiding the token selection with the probabilities that the model already predicts, but masking out tokens that do not align. We believe that this is the cleanest and simplest approach.
>
> 3. We would likely ask the reviewer to rephrase the question. The datasets that we used are only for evaluation. We take publicly pretrained models to do such evaluation.
>
> 4. the effectiveness of token alignment itself is independent of the cross or multi-lingual setting, holding other factors constant. If cross lingual entails rare language, it would require models to have seen sufficient amount of such language to be able to generalize, even with full token scenario. Given such cases where models can output meaningful probability distribution over tokens in each step, our method helps when there is a presence of partial token.
>
> Rare language may entail higher backtracking B. In such case, a tokenizer schema where we know which token corresponds to the beginning of pre-token can be helpful.
>
> 5. Token healing is a similar concurrent approach that is a concurrent work in terms of blog post (not a full paper).
>
> 6. This is a great question. The token alignment is very suitable for 'completion' mode. For instance, if we use language models to help write word document, or write code within the code editor. This is in contrast to the Question - Answer mode where the model does not have to output text that is a continuation of the prompt, but would perhaps start in a different line and start a new sentence / piece of code instead.
>
> 7. Our method works for different input length. In short, if using the model without token alignment can support L input tokens, using token alignment also supports ~ L input tokens as well (give or take the number of backtrack tokens B where we use around 1-3, as demonstrated in the paper)
>
> Thank you again for the thoughtful comments. We will incorporate it in the draft accordingly.

---

### Official Review · Reviewer_kAYh · 2023-10-31

**Soundness:** 2 fair
**Presentation:** 2 fair
**Contribution:** 3 good
**Rating:** 5
**Confidence:** 4

**Summary:**

This paper deals with problems arising from LLM prompts ending with incomplete subword tokens. This is an issue for use cases involving autocompletion, such as code generation. The authors propose a new decoding algorithm to handle partial tokens. The algorithm backtracks to a previous full token, then decodes the subsequent token while limited to only generating tokens that start with the partial token. This enables the model to use next-token predictions for complete tokens within its vocabulary, while ensuring that the resulting model output still aligns with the partially generated user input. The authors outline several classes of common partial token occurrences (e.g. natural language subwords, space prefixes, contiguous spaces in code) and construct evaluation data for testing generation from partial tokens for each class. They show that their backtracking algorithm improves performance across natural language and code generation datasets, with limited increased latency during decoding.

**Strengths:**

**-- 1. The problem of partial tokens is important and neglected –**

The authors have highlighted an important issue with this work. The effect of tokenisation on recent LLMs has not been sufficiently studied, and partial tokens might well be an issue for common LLM use cases like code generation.

**-- 2. Method is simple and effective –**

The token alignment algorithm is an intuitive solution to the studied problem. It is easy to implement for existing LLMs, so it could realistically be used and adapted by practitioners. The reported results also look very good (large gains on some of the datasets), so I do see this work being useful for future research.

**-- 3. Useful categorisation of partial token scenarios –**

The authors outline a categorisation of the different types of common partial token errors. This in itself is a useful contribution, as it highlights the real-world use cases that cause problems.

**Weaknesses:**

**-- 1. Uncertainty around backtracking steps --**

The paper is somewhat unclear about the method of backtracking multiple tokens (B in Algorithm 1), so I would ask the authors to clarify this. The introduction mentions “Our approach involves backtracking to the last complete tokens”, which is how I understood the method initially. But later it is suggested that the method backtracks multiple tokens (B = 3 tokens), and not just one token back.

I am not sure why backtracking 3 tokens would work better than 1, as found in the ablation study. Wouldn’t that give the model less context unnecessarily, given that 1 token back is all we need to get to a complete token?

**-- 2. More details on increased latency --**

The paper would be improved by a more detailed explanation of the increased computational complexity introduced by token alignment. I appreciate that the authors have included a whole subsection on this topic, but I would suggest shifting the focus of the subsection to matters that are practically informative. What is the average added latency, in ms and percentage? This is mentioned, but the authors should include the relevant hardware details for reference as well. Increased latency would be a primary real world concern for practitioners.

**-- 3. Framing of the problem --**

The problem of partial tokens could be presented more accurately in some instances. For example, at the end of page 3 the authors suggest that subword tokens of linguistic words like “banana” (“banan” or “bana”) could lead to issues. I think it would be best to cite work on this, or prove this experimentally? Linguistically unsound subword tokenisations are present in every LLM, yet they work well in most cases, presumably because there is enough data for models to learn how these subwords combine to form words (e.g. on https://platform.openai.com/tokenizer the word “bananas” is segmented “ban-anas”.)

The reported results could be viewed as proof that this is an issue, but for that it would be useful to include more examples of where token alignment helps e.g. can models autocomplete “bana” as “bananas” given sufficient context? Does token alignment allow them to do this?

The same holds for the other partial token categories.

**Questions:**

**-- Questions --**

1. How does your work differ from this paper? https://aclanthology.org/D19-1507.pdf It seems like they are using the same backtracking algorithm (see Section 4), but testing it with different models and datasets.
2. Will you release your evaluation datasets publicly?
3. Section 2, end of paragraph 1. How does subword regularisation increase inference latency?
4. What model’s tokeniser was used for the examples in Figures 1-3? If these examples were constructed by hand, this should be mentioned.

**-- Typos: --**

1. Tables 1-3 have incorrectly boldfaced results that are lower than comparable results (e.g. Table 1, all 3 baselines results for LLaMA with token alignment are incorrectly bold).
2. Section 1, paragraph 3: ’incomplete token “sys” in Figure 2’ -> ’incomplete token “re” in Figure 2’
3. Section 1, paragraph 4 fix unclear wording: “with an average increase of only 3-7 ms for using token alignment, in addition to the number of backtracked tokens”.
4. Section 3, paragraph 2: fix notation “where N is the number of tokens we need to backtrack” -> “where B is the number of tokens we need to backtrack”.
5. Section 3, paragraph 4: “to avoid unnecessary the trie lookup” -> “by avoiding unnecessary trie lookups”.
6. Section 4, paragraph 3: “it is general quite hard” -> “it is generally quite hard”.
7. Fix last sentence on page 4: “the model always obvious contiguous”.
8. Section 5, paragraph 1: fix “for each case described in Section 4 show”
9. Section 5, paragraph 1: “handles the constrain due to all such cases” -> “handles the constraint…”
10. Section 5, paragraph 2: fix unclear wording “processing publicly available datasets to their corresponding variants”.
11. Section 5.2.1, paragraph 3: “token alignment can be use in all cases” -> “t token alignment can be used in all cases”.
12. Section 6.2, paragraph 2: fix unclear wording “opportunities to mark some token as beginning of pretoken as building time”.

---

> ### Author Response · Authors · 2023-11-23
> **Response to Reviewer kAYh**
>
> Thank you for your detailed comments and feedback. It is clear that the reviewer understands the paper deeply and have spent time to read our work -- we truly appreciate it!!
>
> Addressing Strengths:
> - Thank you for acknowledging that the problem of partial token is important but largely neglected due to the artifact of most evaluation setup.
> - Thank you for recognizing the effort on categorizing the partial token scenarios. Some scenarios are highly not obvious and we make effort to select out these cases for surgical studies of the language models' behavior.
>
> Addressing Weaknesses:
> 1. This is a very good question that we will emphasize further to make it clear! For different tokenizers, it can be the case that there can be no 'marker' if that token is the start of a pre token. In this case, B needs to be pre-set to be a certain number. However, in some tokenizer, such as SentencePiece, we can have such marker for pre-token, which allows us to backtrack with different B dynamically. (B can be 1 for some input and 2 or other inputs)
>
> Backtracking 3 tokens can be better in certain case. For example, if we have a very long token that gets broken up into partial tokens. For instance, if the full token is "abcdefghijk" (synthetic example here) but the prompt is "abcdefgh" -- instead of being a full token, this prompt likely gets broken up into something like 'abcd', 'ef', 'gh'. If we do not roll back for 3 steps in this case, then the prompt that the model sees still corresponds to partial tokens that got broken up. This would still constrain the model due to the artifact of tokenization.
>
> However, if we correctly rolls back to before "abcdefgh", then the model can see that the token "abcdefghijk" matches in prefix with the text "abcdefgh" -- so our token alignment method will do the job correctly.
>
> That being said, B=3 typically is not required for most prompts, but it does not hurt to roll back in order to a full token that may be broken into up to 3 tokens as well. However, as the reviewer nicely pointed out, it gives the model less context. Therefore, it is a balance between giving the model context and precise alignment. The alignment process in itself may be seen as guiding the model in a way, however.
>
> 2. This is a good point. We initially focus on the number of additional tokens since it will hold even though in the future the decoding process itself can be much faster, relative to the input processing latency etc. For instance, speculative decoding can make the latency per token much faster, which results different additional latency (lower) compared to without using speculative decoding.
>
> For instance, on A100 GPU, for a 2000-token context input takes around ~400 ms for a moderate size model (7B LlaMA for instance). Generating 256 tokens without speculative decoding can take 256*30 ms. The overhead of token alignment with B=3 will be around 3*30 ms. In total, the overhead is 90/(400 + 256*30) = 1.1%
>
> 3. Thank you for pointing this out. We will add additional examples to make it clear. For the current draft, examples given in Figure 1, 2, 3 and in Section 4 illustrates various scenarios. For example, in figure 3, "_ _ _" + "_ return" are the fully tokens corresponding to 4 spaces + "return". In the scenario where the model is presented to " _ _ _ _" as the end of a prompt, then it constitutes a partial token issue.
>
> It is true that unsound subword tokenization are prevalent. However, once the prompt breaks from the patterns that the models observe during training, then problem arises. Such discrepancies happens when the end of the prompt is cut off before a full token.
> For instance, 'banana' could be composed of two tokens 'ban' and 'ana'. If the prompt ends with "monkeys eating yellow ban", the the most likely next token is 'ana' and there is nothing preventing the model to predict such a token.
>
> However, if the prompt ends with "monkeys eating yellow bana" and the models have seen 'ban' + 'ana' for every semantic instance of 'banana' during training, then the model will most likely not predict 'na' after this prompt.
>
> Addressing questions:
> 1. Thank you for pointing out this work.  The paper mentioned uses beam search together with approximate marginalization. For modern language models, beam search is not typically used but popular methods are top-p (nucleus sampling) and top-k. Our proposed approach is designed to be compatible with top-p and top-k and works seamlessly with such sampling approaches. We will add this to related work and contrast the difference, however. Thank you again!
>
> 2. Yes.
> 3. Subword regularization breaks up a piece of text into more tokens and show this to the model during training time. The model then has higher chance of using higher number of tokens to produce the same text, compared to without subword regularization.
> 4. These are constructed by hands. We will point it out.
>
> Thank you for pointing out the typos. We are extremely grateful for the detailed list!

---

> > ### Comment · Reviewer_kAYh · 2023-11-23
> >
> > Thank you for responding to my queries. Including your clarifications in the paper will certainly strengthen the work. Your comments clarifying the multi-step backtracking are especially useful. I have increased my overall rating of the paper, as well as the contribution score.
> >
> > A few comments on your response:
> >
> > **-- Weakness #3: framing the problem of partial tokens --**
> >
> > I understand your reasoning around this, but I still feel it would be worth further exploring the problem itself to a greater extent.
> >
> > > However, if the prompt ends with "monkeys eating yellow bana" and the models have seen 'ban' + 'ana' for every semantic instance of 'banana' during training, then the model will most likely not predict 'na' after this prompt.
> >
> > This could be true, but LLMs trained on enough data are also exposed to a lot of noise in the training data (e.g. spelling variations, misspellings, incorrect spacings) that they could potentially learn to deal with something like a partial token. Previous work (https://arxiv.org/pdf/2206.02608.pdf) has shown that some models actually learn which characters their tokens consist of. Such knowledge could enable models to still work with partial tokens.
> >
> > The contributions of the work would be stronger if it included an exploration of the problem before trying to fix it.
> >
> > **-- Subword regularisation latency --**
> >
> > Depending on the subword regulariser, they do not necessarily introduce more tokens. For example, sampling from ULM could lead to less/more segmentation, depending on the sample.

---

### Official Review · Reviewer_fPS1 · 2023-10-31

**Soundness:** 3 good
**Presentation:** 3 good
**Contribution:** 2 fair
**Rating:** 6
**Confidence:** 3

**Summary:**

The authors propose a method for text completion in large language models (LLM) from incomplete tokens in prompts. The method uses an algorithm to backtrack generated tokens for the completion of sub-words.  The main contributions are: i) method for identification and processing of incomplete tokens, and ii) comparison of the proposed method with byte-pair-encoding (BPE) baselines on code and natural language processing (NLP) benchmarks. The proposed method shows competitive results compared to the baseline.

**Strengths:**

- The proposed method  tackles issues with the text completion in LLM with a small latency overhead.
- Clear description of the proposed approach.
- The authors perform a comparison of the proposed approach on code and NLP benchmarks.

**Weaknesses:**

- It is not clearly described the background knowledge needed to motivate and position the proposed method in the literature.
- It is not clearly described the alignment task and the relation to code and NLP.
- A possible extra contribution can be the addition of a statistical significant test of the results.

**Questions:**

Please address the following questions during the rebuttal:

- Please elaborate in the background used to develop the proposed method.
- Is the alignment task based on fine-tuning a model or in-context-learning? Could you elaborate more on differences and benefits, compared to your method.
- Could you elaborate on the selection and importance of hyper-parameters? such as backtracking B.
Is the selected baseline a strong proposal compared to other related work on fine/instruction tuning or character-based models?

Extra:

Please add related-work/literature context to the introduction and methodology sections

**Details Of Ethics Concerns:**

I have no concerns.

---

> ### Author Response · Authors · 2023-11-23
> **Response to Reviewer fPS1**
>
> We thank reviewer for your time and for the questions/comments.
>
> - Thank you for acknowledging the strengths of the paper.
> Addressing Weakness comments:
> - We describe the motivation in the introduction as well as accompanying figures where if prompt gets tokenized where the last token corresponds to a partial token, this corresponds to mismatch between training time versus inference time. The scores where we use token alignment versus without using demonstrates that partial token can hurt language model's generation abilities significantly.
> - The token alignment task is applicable to *any* text completion task -- we illustrate examples in Figure 1 and 3 and provide experiments for both text and code throughout the paper.
> - We make note on the statistical significant -- due to the scale of the evaluation we do not have them by default in this draft.
>
> Addressing questions:
> - The token alignment is applicable for both pretrained and finetuned model. In our case, we use pretrained model. We also experimented with few-shot prompting which is the 'in-context learning' scenario (Table 3).
> - We experimented with different values of B (number of backtrack tokens) -- Table 6 in appendix illustrates our extensive result. For the main paper, we use B=3. For B=1 or 2, we observe small differences.
>
> We will aim to clarify the paper further based on the suggestions.

---

> > ### Comment · Reviewer_fPS1 · 2023-11-23
> >
> > Thank you for addressing my questions. I have no further comments.

---

### Official Review · Reviewer_dd8B · 2023-11-01

**Soundness:** 2 fair
**Presentation:** 2 fair
**Contribution:** 1 poor
**Rating:** 3
**Confidence:** 4

**Summary:**

The paper proposes a token alignment model to apply in auto-completion task. The proposed method is described well and tested in the code generation task. There are some improvements recommended due to connection to literature and showing evidence that there is a problem that the paper addresses beyond the application of the LMs in code generation. The authors are referred to decoding literature in LM and sequence models in previous tasks for a fair comparison.

**Strengths:**

Interesting approach to improve efficiency in code generation.

**Weaknesses:**

The main problem of this paper is that there is no evidence or support that this problem exists, and caused by subword misalignment.
Related work do not exactly align with the problem studied and there is no discussion why authors do not find evidence on the problem they choose to address.
Novelty of the proposed model is not clear.

**Questions:**

Can the authors find or add in their study evidence through actual experiments on the problem they propose to solve?
There are many variants of beam search that could do hierarchical subword/word generation. How do this approach differ from the existing methods?

---

> ### Author Response · Authors · 2023-11-22
> **Response to Reviewer dd8B**
>
> We thank you for your comment and questions.
>
> Regarding the necessity of token alignment, we have numerous results showing that without token alignment, when prompt ends with partial token in all the four scenarios we categorized (subword, punctuation, white spaces due to space prefix and contiguous spaces), the drops dropped drastically without a method such as token alignment to help alleviate the tokenization artifact.
>
> We also have provided illustrations in Figure 2 and Figure 3.
>
> A concurrent work in terms of blog post also studies this exact topic, for instance, where our method is similar. (we cited this blog post in the paper as well under Related Work)
>
> Our work do not use beam search and is designed to work with the existing approach of incremental decoding in language model which is nucleus (top-p) sampling as well as top-k.
>
> Prompt boundaries and token healing, 2023. URL https://towardsdatascience.com/the-art-of-prompt-design-prompt-boundaries-and-token-healing-3b2448b0be38

---

### Author Response · Authors · 2023-11-23
**Response to Reviewers**

We thank reviewers for thoughtful comments and questions. We emphasize our contribution below:

As reviewer kAYh pointed out, the problem of partial token for language models is important but is often neglected. The issue becomes obvious in the case of text *completion*, a scenario where we generate text as a continuation of the prompt. (rather than question-answer). This is of important practical application, such as editing or completion mode where we use language model to edit paragraphs, or write code within our integrated development environment (IDE). Hence, we demonstrate that our simple and clean approach can *uniformly handle all partial token scenarios*.

We have categorized subtle but important scenarios where partial token and the artifact of tokenization is the culprit. We *develop datasets* dedicated to such scenarios in order to individually understand these cases, which highlights the benefits of token alignment. For instance, without token alignment, partial token can degrade code execution scores by more than 10 points (all scenarios in Python, scores dropped by at least 10 points without token alignment).


We have incorporated comments and addressed questions in a new pdf draft. We edited the main text and as well as the FAQs section in the appendix where we have included representative questions from reviewers. We believe this paper is highly suitable for ICLR based on the impact of this method for real-world language model usage and encourage users to adjust the scores due to such potential impact.

---

### Meta-Review · Area_Chair_StPX · 2023-12-06

**Metareview:**

The paper studies the problem of language models failing to correspond correctly to prompts which end in (potentially) partial word tokens. A backtracking method is proposed to deal with this issue. Results show performance improvements across various tokenization setups in prompts ending with partial tokens. A disadvantage is that the method increases latency. The main weaknesses of the paper are that the framing of the problem is not precise enough and that there isn't a strong enough demonstration that it is a problem in practice. The technical novelty of the paper is not clear enough and there is insufficient comparisons to alternative solutions to the problem.

**Justification For Why Not Higher Score:**

The paper identifies and addresses a real practical issue in LLMs, but the problem is not motivated rigourously enough, and it is unclear if there is strong enough technical contribution.

**Justification For Why Not Lower Score:**

N/A

---

### Decision · Program_Chairs · 2024-01-16

Reject